# Characterizing the Relationship Between Intervention Delivery and Outcomes Within Part C Community Settings

**DOI:** 10.3390/bs15101394

**Published:** 2025-10-15

**Authors:** Katherine Pickard, Scott Gillespie, Aubyn Stahmer, Jennifer Singh, Lawrence Scahill

**Affiliations:** 1Department of Pediatrics, Emory University, Atlanta, GA 30322, USA; scott.gillespie@emory.edu (S.G.); lawrence.scahill@emory.edu (L.S.); 2Children’s Healthcare of Atlanta, Atlanta, GA 30342, USA; 3MIND Institute, University of California, Davis, CA 95616, USA; astahmer@ucdavis.edu; 4School of History and Sociology, Georgia Institute of Technology, Atlanta, GA 30332, USA; jennifer.singh@hsoc.gatech.edu

**Keywords:** autism, early intervention, implementation, adaptation, parent-mediated intervention

## Abstract

Routine Early Intervention services are an ideal context to evaluate parent-mediated intervention (PMI) delivery. While effectiveness research suggests that receiving manualized PMIs positively affects caregivers’ learning and use of intervention strategies, the impact of other aspects of delivery, such as PMI adaptation, on caregiver engagement and learning is less clear. The current study aimed to address this gap by closely characterizing the delivery and associated outcomes of an autism PMI, Project ImPACT, within an Early Intervention (EI) Part C system. In total, 21 EI providers and 23 caregivers of children with social communication delays participated. Following training in Project ImPACT, the providers submitted videos of their Project ImPACT sessions as part of routine service delivery. The sessions were behaviorally coded for Project ImPACT coaching fidelity and instances in which Project ImPACT was adapted. After each session, the caregivers rated their participatory engagement and therapeutic alliance. Before and immediately following the intervention, the caregivers also completed measures of their self-efficacy and their child’s social communication skills, and their use of Project ImPACT strategies (i.e., fidelity) was behaviorally coded. The results demonstrated that EI providers’ Project ImPACT coaching fidelity was not related to caregiver ratings of therapeutic alliance or participatory engagement at the session level. Augmenting Project ImPACT sessions was associated with higher caregiver ratings of therapeutic alliance but not with participatory engagement. Although provider coaching fidelity was not associated with changes in caregiver ratings of self-efficacy, it was associated with caregiver use of Project ImPACT strategies focused on teaching their children new skills. There was no association between provider fidelity and caregiver report of child social communication outcomes. The current study highlights the complicated relationship between the delivery of autism PMIs and caregiver-reported outcomes. The findings highlight the value of holistic delivery models that support adaptations in response to child- and family-level factors.

## 1. Introduction

Parent-mediated interventions (PMIs) are an early intervention approach that empowers parents to learn and use strategies that promote their child’s development ([5]; [20]; [21]). PMIs operate under the assumption that, with the support of a provider, caregivers learn and use empirically supported intervention strategies within home and community routines. Caregiver use of intervention strategies is presumed to support the development of their child’s social communication and daily living skills (i.e., [7]). Within tightly controlled trials of PMIs, caregivers experience significant increases in their self-efficacy ([26]; [10]) and increase their use of core intervention strategies within play and daily routines ([12]). Further, caregivers’ use of intervention strategies (i.e., their fidelity) mediates their child’s social communication outcomes, suggesting that PMIs exert their influence via their impact on caregiver strategy use ([34]; [12]).

Although PMIs are well-aligned for implementation within community Early Intervention settings, their multi-level and multi-component format may make them complicated to deliver by non-specialty providers ([7]; [28]). For example, PMIs require that providers effectively explain and model intervention strategies, provide live feedback to caregivers during their practice of strategies, and facilitate caregiver reflection and practice planning ([7]; [23]). In addition to the mastery of the coaching processes, PMIs require that providers understand and effectively communicate the technicalities of specific intervention strategies that are the focal point of coaching. For example, providers might need to explain the rationale and key elements of strategies such as following a child’s lead, modulating affect to support engagement and regulation, and using motivating activities to encourage and reinforce the use of new communication skills ([30]).

In the absence of formal training in PMIs, EI providers are inconsistent in their use of general parent coaching strategies, such as the use of demonstration, in vivo feedback, and practice planning (e.g., [6]; [22]). Even after participating in ongoing PMI training and consultation, there is considerable variability in how EI providers coach caregivers to implement intervention strategies as part of both effectiveness research (e.g., [27]; [32]) as well as research conducted as part of ongoing contracts to train EI providers within Part C systems (e.g., [25]). The inconsistency in how providers deliver PMIs may be driven by EI providers’ diverse disciplinary backgrounds and skillsets (e.g., [1]; [9]), the complexity of PMI models, as well as providers’ attempts to tailor the delivery of manualized PMIs to fit the EI service context ([25]).

Given the variability in PMI delivery, EI systems are an ideal setting in which to unpack the associations between providers’ routine PMI delivery and caregiver outcomes. To date, research evaluating these relationships has predominantly occurred within the context of effectiveness research. Within this context, caregivers who receive PMIs from trained EI providers have greater intervention strategy use at the group level than caregivers who receive early intervention services from untrained providers ([27]). Other research evaluating fidelity–outcome relationships more directly has shown that EI providers’ coaching fidelity is associated with caregivers’ use of responsive parenting strategies but not with their use of specific PMI intervention techniques ([32]). In these studies, children who participated in manualized PMIs had greater social communication gains than children receiving services from untrained providers. It was not clear, however, how an EI providers’ coaching fidelity related to child outcomes.

In addition to lingering questions regarding the impact of EI providers’ coaching fidelity, there is also much to be learned about how other aspects of PMI delivery—including adaptation—affects caregiver outcomes. By and large, community clinicians perceive that adaptations to PMI enhance their fit for diverse patient populations ([14]; [17]; [29]) and are necessary when working within underrepresented community contexts ([2]). Within the EI context, providers adapt manualized PMIs in greater than half of the sessions. While providers who spend more time tailoring their PMI approach have reduced coaching fidelity in the same session ([25]), we do not know how these same adaptations affect caregiver outcomes. Given that EI providers often cite maintaining caregiver engagement (i.e., caregiver understanding of and motivation to participate in therapeutic sessions) and therapeutic alliance (i.e., strength of the relationship between therapist and caregiver) as a primary reason to modify their PMI approach (e.g., [24]; [29]), unpacking how adaptations affect caregiver outcomes is essential. 

To date, limited empirical work has been conducted to evaluate how aspects of PMI delivery, including both fidelity and adaptation, affect caregiver engagement and learning within EI systems. Without this understanding, it is difficult to make informed decisions about how to effectively balance PMI fidelity with the flexibility necessary to deliver manualized programs within EI and other community settings. In response, the overarching goal of this study was to demonstrate how fine-grained data collected from each session can be used to evaluate the association between EI providers’ delivery of an evidence-based autism PMI called Project ImPACT ([11]) and caregiver outcomes as part of routine implementation. The specific questions include the following: (1) How do providers’ Project ImPACT fidelity and adaptation affect caregivers’ reported participatory engagement and therapeutic alliance? (2) How do providers’ Project ImPACT fidelity and adaptation delivery affect changes in: (a) caregiver self-efficacy; (b) caregivers’ use of Project ImPACT strategies; (c) child social communication outcomes?

## 2. Materials and Methods

### 2.1. Setting

This project was embedded within an ongoing training contract with Georgia’s Department of Public Health (DPH). Georgia’s DPH oversees the state’s Part C Early Intervention system, called Babies Can’t Wait (BCW). Babies Can’t Wait serves approximately 19,000 children under the age of three each year, approximately 2500 of whom screen as having an increased likelihood of being autistic. Each year, Project ImPACT training is offered to EI providers as part of ongoing contractual arrangements. The providers who enroll in training can deliver Project ImPACT within routine EI services and are also given the option to participate in research alongside training.

### 2.2. Study Design

This study sought to characterize the delivery and associated caregiver outcomes of Project ImPACT as part of routine implementation practice. The study procedures were approved by the Institutional Review Board at Emory University and the Georgia Department of Public Health. Data collection included a combination of observational and survey data that was collected from EI providers and caregivers alongside the delivery of Project ImPACT.

### 2.3. Inclusion Criteria

Providers were eligible for research participation if they satisfied the following criteria: (1) were currently employed or contracted by Georgia’s EI system; (2) maintained an active therapy caseload, as defined by seeing at least one child 12–34 months of age with social communication delays as measured by a positive screen on the Parent Observation of Social Interaction (POSI). The POSI is a 7-item measure that asks caregivers to rate their child’s use of social communication skills, including communication, gesturing, and play. It has a has a sensitivity of 93.6% ([31]). Providers were required to see eligible children at least 2 h per month. Given that the study occurred as part of routine training initiatives, EI providers were not randomized. State-level EI leadership supported the recruitment of participants by disseminating training and research flyers through provider listservs used across the EI system. Providers were able to contact the study team to express interest in Project ImPACT training participation. All EI providers who enrolled in training and met the eligibility criteria were trained in Project ImPACT and able to opt into research procedures (e.g., additional survey, observational, and qualitative measures) alongside Project ImPACT training to evaluate routine delivery and adaptation.

The participating providers were encouraged to share caregiver recruitment materials with the participating families of children with social communication delays on their caseload using IRB-approved flyers and/or a video. Interested caregivers were then able to complete a brief interest survey that provided their contact information to the research team. Caregivers were eligible to participate if they satisfied the following criteria: (1) had a child under 36 months currently enrolled in services through Georgia’s Early Intervention system; (2) were being seen by a provider participating in Project ImPACT training and research; (3) their child screened positive on the Parent Observation of Social Interaction (POSI). Caregivers received a USD 25 gift card for completing the research surveys before and after receiving Project ImPACT. 

### 2.4. Provider Participants

Forty-one EI providers enrolled in routine Project ImPACT training. Of those, 36 providers completed the training, and 26 providers met the inclusion criteria for the current study. The most common reason for not meeting the inclusion criteria was not carrying an active therapy caseload due to being in a service coordinator role. Of those trained and eligible, 21 providers participated in the current study. All providers identified as female and reported M = 8.61 years of experience working within EI systems (*SD* = 7.2). See Table 1 for provider demographic information.

### 2.5. Caregiver Participants

Twenty-seven caregivers expressed interest in participating in the current study and twenty-three enrolled alongside their EI provider. All caregivers identified as mothers and had children currently enrolled in Part C services. In total, 52% of the caregivers identified as Black or African American, and 13% as Hispanic or Latina. The children were 60.9% male and M = 24.7 months old (*SD* = 5.7 months). All children screened positive for social communication concerns on the Parent Observation of Social Interaction (POSI; [31]). See Table 2 for caregiver and child demographic information.

### 2.6. Intervention

Project ImPACT is a manualized PMI that supports social communication skills in young children ([11]). The providers delivering Project ImPACT coach caregivers to use a blend of developmental and naturalistic behavioral intervention techniques across daily routines to support their child’s social engagement, communication, imitation, and play skills. Project ImPACT can be delivered once each week for one hour over the course of 12 to 24 weeks in both in-person and telehealth delivery models. The program begins with collaborative goal setting with caregivers in social communication skill areas targeted by the program. In subsequent sessions, the caregivers receive the following: (1) didactic instruction in intervention strategies; (2) modeling of the intervention techniques; (3) live coaching while practicing with their child; (4) a practice plan for implementing the targeted strategies with their child in a daily routine or activity. 

### 2.7. Intervention Training and Consultation 

EI providers first completed an asynchronous 6 h online tutorial that offers an overview of Project ImPACT, followed by a four-part live webinar series conducted via Zoom. The 14 h webinar series includes didactic instruction while allowing for Project ImPACT role play, video review, and implementation planning. Following completion of the online tutorial and webinar series, the providers participated in group consultation for one hour per week across approximately 12 weeks. Group consultation was structured such that it embedded a combination of didactic instruction, role play to facilitate behavioral rehearsal, video review of provider sessions, joint problem solving, and planning around subsequent sessions. As part of the consultation, the participating EI providers submitted videos of each of their Project ImPACT sessions with their participating caregiver(s) which were scored for Project ImPACT coaching fidelity and Project ImPACT adaptation (see below).

### 2.8. Measures

The measures for the current study included a combination of caregiver-report measures, as well as observational measures. The measures included those that occurred immediately prior to and at the conclusion of receiving Project ImPACT, as well as measures completed at the end of each intervention session. See Table 3 for the measure timeline for this study.

#### 2.8.1. Caregiver Measures

Demographic information. The caregivers provided demographic information about themselves and their child. This included their age, race, ethnicity, family size, marital status, and highest level of education obtained. The caregivers also indicated the age, sex, race, and ethnicity of their child receiving the EI services, as well as their child’s developmental and/or medical diagnoses.

Session participatory engagement. At the conclusion of each Project ImPACT session, the caregivers were asked to complete the Parent Participatory Engagement Measure (PPEM), an 11-item caregiver-report measure of in-session participation within PMIs for autism. In the current study, two sub-scales were calculated, as follows: (1) receipt of coaching, in which the caregivers rated the extent to which their provider demonstrated the intervention strategies, coached the caregiver to use the intervention strategies, and developed a practice plan (e.g., “in today’s session, how much did your therapist show you how to use a new skill?”; (2) caregiver engagement, in which the caregivers indicated their question asking, comfort speaking up, and perception of practice plan within the session (e.g., “during today’s session, how often did you ask your child’s therapist questions?”). The caregivers rated each item using a 5-point scale. The PPEM demonstrated excellent internal consistency (alpha 0.86–0.93) across multiple studies.

Session therapeutic alliance. After each session, the caregivers completed the 5-item Enabling and Partnership sub-scale of the Measure of Processes of Care (MPOC-20; [15]). The caregivers rated their agreement with each of the five items using a 5-point scale. Example items included “my therapist made sure I got to say what was important to me.” For each session, an average therapeutic alliance score was calculated by summing item ratings and dividing by five.

Caregiver Project ImPACT fidelity. A 10 min parent–child interaction was recorded as part of the first and final Project ImPACT session within the context of play and/or daily routines. These interactions behaviorally coded specific caregiver behaviors, including the following: (1) following the child’s lead; (2) adjusting communication; (3) creating opportunities for initiation; (4) teaching new skills; (5) shaping the interaction. All strategies were scored using a 5-point scale, where 1 indicated that the caregiver did not use the strategy, and 5 indicated that the caregiver used the strategy consistently. Reliability was scored on 20% of the sessions by trained clinicians and was excellent (ICC = 0.96). 

Family Outcome Scale, Revised (FOS-R; [3]). The FOS-R comprises 24 items that assess empowerment-related constructs expected to change through participation in early intervention programs, including Part C EI systems. It comprises items across five outcomes: (1) understanding your child’s strengths, needs, and abilities; (2) knowing your rights and advocating for your child; (3) helping your child develop and learn; (4) having support systems; (5) accessing the community. The caregivers rated each of the 24 items using a 5-point scale. Sub-scale scores were calculated as the sum of each item score within the sub-scale.

Social Communication Checklist. (SCC; [11]). The SCC was developed as part of Project ImPACT to support the creation of developmentally appropriate social communication goals to guide program delivery. The SCC is a 70-item checklist in which parents indicate if a child uses each skill “rarely/not yet (1),” “sometimes, but not consistently (2),” or “usually, at least 75% of the time (3).” The items are listed in a developmental sequence and grouped such that they correspond with the social communication domains targeted in Project ImPACT: 15 social engagement items, 15 expressive communication form items, 15 expressive communication function items, 8 receptive language items, 6 imitation items, and 11 play items. The item scores were summed for domain scores.

#### 2.8.2. Provider Measures

Provider demographic information. Prior to training participation, the providers reported their age in years, gender identity, race, ethnicity, and educational attainment. They also provided information on their professional discipline, employment status (i.e., contracted or employed provider), years of experience working within EI systems, years of experience working with autistic children, and their current caseload size. 

Project ImPACT coaching fidelity. The recorded Project ImPACT sessions were behaviorally coded using the Project ImPACT Coaching Fidelity Checklist ([11]). The checklist includes items about: (1) setting up the coaching environment and preparing session materials; (2) using coaching strategies (i.e., reviewing the practice; introducing the session topic, demonstrating strategies; supporting caregiver practice of strategies; creating a plan for the practice); (3) using collaborative strategies to partner with the caregivers. Each of the 21 items was rated on a three-point scale where a score of 1 indicated that the coaching behavior was not observed, and a score of 3 indicated that the coaching behavior was fully observed. Total fidelity scores for each session were summed up and divided by the total possible fidelity points to achieve a percent rating (0–100%). Fidelity was coded by two graduate-level clinicians certified in Project ImPACT. Reliability was calculated on 20% of all recorded sessions. The average intraclass correlations (ICCs) for individual fidelity items was 0.77. 

Project ImPACT adaptation. Each Project ImPACT session was behaviorally coded for the presence of adaptations using the Framework for Reporting Adaptations and Modifications to Evidence-Based Interventions (FRAME-IS; [33]). Based on this frameworks and prior research (e.g., [17]), adaptations were grouped into those in which Project ImPACT was augmented (i.e., core Project ImPACT elements were supplemented or extended) and those in which Project ImPACT was reduced (i.e., in a manner that might attenuate the impact of core Project ImPACT functions). Examples of augmenting adaptations include repeating Project ImPACT topics, lengthening Project ImPACT sessions, and integrating other therapeutic content or approaches, such as integrating therapeutic strategies related to toilet training, sensory regulation, or service navigation. Examples of reducing adaptations include dropping Project ImPACT coaching activities (e.g., dropping coaching, demonstration, and/or practice planning) and skipping Project ImPACT topics. Each adaptation was rated as present or absent by the same trained individual coding Project ImPACT fidelity. Reliability was calculated on 20% of the sessions and indicated 81% agreement.

### 2.9. Analytic Approach

Descriptive statistics included frequencies and percentages for categorical variables and means and standard deviations for continuous variables.

To account for repeated observations within caregiver–provider dyads, linear mixed models were used for both correlational and group comparison analyses. Random effects were included, where appropriate, to address the clustering of sessions within and between dyads. Repeated measures correlations were estimated using the SAS macro MMCorr_NormalApprox ([13]), based on the method described by [8] ([8]). These models produced correlation estimates with normal approximated 95% confidence intervals (CIs) and *p*-values. Least-squares means and corresponding 95% CIs and *p*-values were reported for group comparisons.

Paired *t*-tests were used to evaluate changes over time (e.g., from Time 1 at the start of the intervention to Time 2 at completion). Partial Pearson correlations were used to examine associations between aggregated provider coaching fidelity and Time 2 outcomes, controlling for baseline (Time 1) values. When applicable, effect sizes were calculated to supplement the *p*-values. For mixed model group comparisons, effect sizes were computed by dividing the least-squares mean difference by the square root of the average residual variance from the mixed models. For paired *t*-tests, effect sizes were calculated by dividing the mean difference by the standard deviation of the paired differences. Effect sizes were interpreted using Cohen’s d criteria: small (0.2), moderate (0.5), and large (0.8). Aggregated coaching fidelity scores were computed by averaging the fidelity ratings across multiple sessions for each caregiver–provider dyad. All analyses were conducted using SAS version 9.4 (SAS Institute Inc., Cary, NC, USA). Statistical significance was evaluated using two-sided tests with an alpha level of 0.05.

## 3. Results

### 3.1. How Do Providers’ Project ImPACT Fidelity and Adaptation Affect Caregivers’ Reported Participatory Engagement and Therapeutic Alliance?

As depicted in Table 4, Project ImPACT coaching fidelity was not significantly associated with caregiver ratings of therapeutic alliance (*p* = 0.99) or participatory engagement (*p* = 0.18) across up to 60 intervention sessions.

As depicted in Table 5, Project ImPACT sessions with an augmenting adaptation had significantly higher caregiver ratings of therapeutic alliance than those without an augmenting adaptation. However, the sessions with augmenting adaptations did not have significantly higher ratings of participatory engagement. Project ImPACT sessions with reducing adaptations did not differ from those without reducing adaptations in terms of therapeutic alliance or participatory engagement. 

### 3.2. How Do Providers’ Project ImPACT Fidelity and Adaptation Affect Caregiver Self-Efficacy; Caregivers’ Use of Project ImPACT Strategies; And Child Social Communication Outcomes?

Table 6 evaluates changes in caregiver knowledge and empowerment, caregiver fidelity, and child social communication skills from the initiation of Project ImPACT through program completion. Significant improvements were observed in the Understanding Strengths and Needs and Helping Your Child Develop and Learn domains of the Family Outcomes Scale, Revised (FOS-R), as well as in nearly all domains of Project ImPACT caregiver fidelity and all domains of child social communication skills.

EI providers’ average coaching fidelity was not significantly associated with the Time 2 scores on the Family Outcomes Scale, Revised (FOS-R). However, the direction and magnitude of the association suggested a potential trend, with higher coaching fidelity possibly linked to lower caregiver confidence in supporting their child’s development. While providers’ average coaching fidelity was not significantly associated with caregivers’ use of Project ImPACT strategies after controlling for Time 1 use, the magnitude and direction of the association suggested a positive trend. Specifically, higher coaching fidelity appeared to be linked with greater caregiver use of strategies to create opportunities for their child and to teach new skills. Providers’ coaching fidelity was not associated with caregivers’ ratings of their child’s social communication outcomes (see Table 7).

Finally, we used repeated measures correlations between caregivers’ Project ImPACT fidelity to developmental and naturalistic behavioral strategies and each of the domains of the social communication checklist. As can be seen in Table 8, caregivers’ fidelity to developmental strategy domains (i.e., Focus on Your Child and Adjust Communication) and behavioral strategy domains (i.e., Create Opportunities and Teach New Skills) was not associated with child social communication outcomes. 

## 4. Discussion

Community service settings are an ideal context in which to evaluate how autism PMI delivery relates to caregiver and child outcomes. While previous research has shown that PMI coaching fidelity is associated with caregivers’ use of responsive practices (e.g., [32]), it was not clear how other aspects of PMI delivery influence a range of caregiver outcomes. The current study sought to address this gap by evaluating associations between Project ImPACT coaching fidelity, Project ImPACT adaptation, caregiver engagement, caregiver fidelity, and child social communication outcomes. Given the small sample size, our goal was to highlight methods that could be used to unpack these relationships, as well as preliminary associations. Results from 23 caregiver–child dyads suggested that EI providers’ Project ImPACT coaching fidelity was not associated with caregiver ratings of therapeutic alliance or participatory engagement. However, integrating other topics into Project ImPACT (i.e., an augmenting adaptation) was associated with higher caregiver ratings of therapeutic alliance within the same session. Although provider coaching fidelity was not associated with changes in caregiver ratings of self-efficacy, it was positively associated with caregivers’ use of Project ImPACT strategies focused on teaching children new skills. There was no association between provider fidelity and child social communication outcomes. 

The current study highlights the complicated relationship between PMI delivery and service outcomes and has several implications for autism service delivery within community settings. First, our study findings are consistent with previous research showing positive associations between EI providers’ coaching fidelity and caregivers’ use of Project ImPACT strategies within play and daily routines ([32]). Interestingly, in the current study, EI providers’ coaching fidelity was only associated with caregivers’ use of Project ImPACT strategies focused on creating opportunities and encouraging children to use new skills. It was not associated with caregivers’ use of developmental strategies focused on following their child’s lead or modeling and expanding communication. It may be the case that caregivers come into PMIs with more familiarity and use of foundational developmental strategies ([16]) such that a providers’ coaching fidelity matters less to teach developmental skills. However, strategies to effectively encourage new skills might be novel for caregivers ([16]), and thus, more robust coaching skills are necessary to effectively transfer these strategies to caregivers. If true, these findings highlight the importance of being able to effectively pace the delivery of manualized PMIs by speeding up or slowing down teaching based on caregivers’ familiarity with and use of certain strategies.

Within the current study, the caregivers demonstrated significant increases in their self-efficacy over the course of participating in Project ImPACT. However, their EI providers’ coaching fidelity was negatively associated with these changes. While previous research has shown that caregivers feel more confident following participation in PMI (e.g., [10]), it is possible that significant changes in caregiver self-efficacy are driven by factors other than provider fidelity, such as caregivers’ actual learning and use of therapeutic strategies or how complex their child’s support needs are. It is also possible that caregivers’ feelings of self-efficacy are driven by relational aspects of PMI fidelity (e.g., using a strength-based approach; demonstrating effective partnership) rather than the use of all coaching components (e.g., didactic instruction in therapeutic strategies) or the fact that providers have higher fidelity if perceiving that caregivers have low self-efficacy. Given the small sample size, we could not tease apart whether more specific aspects of fidelity or other factors might drive changes in caregiver self-efficacy; however, this is an important area of future research.

The current study used fine-grained data collection that, while intensive, helped to understand the relative effect of both PMI fidelity and frequent adaptation that is the norm within community settings (e.g., [18]). Within previous research, EI providers have reported adding topics to PMIs and repeating session topics to maintain caregiver engagement and rapport ([25]). While preliminary, the findings from the current study suggest that adding therapeutic topics and/or repeating Project ImPACT sessions may indeed support caregiver perceptions of therapeutic alliance within the same session. This is an important finding, given that therapeutic alliance can be a strong predictor of attendance and therapeutic outcomes across psychosocial interventions (e.g., [4]; [19]). At the same time adapting Project ImPACT was not associated with caregiver ratings of participatory engagement. This suggests that following caregivers’ lead to incorporate other therapeutic content may make caregivers feel heard and more aligned with their providers but may not increase their active participation within therapy sessions.

Given the importance of both EI provider coaching fidelity and adaptation, the findings from the current study highlight the importance of more explicit guidance to EI providers on PMI delivery that balances adaptation with core intervention content. Explicitly acknowledging this balance and teaching clinical decision making around adaptations may help to align manualized PMIs with public EI systems. In order to provide effective guidance to providers within the context of community training initiatives, it will be important to understand both EI providers’ and caregivers’ perspectives regarding the decision-making processes that are needed to drive the effective delivery and adaptation of manualized PMIs. For example, specific considerations that might influence the delivery of manualized PMIs could be the timing and intensity of discipline-specific goals (e.g., fine motor skills), the family’s preferences, the extent to which other service providers are available to address distinct goal areas, the amount of time that the provider will be working with the family, and the intensity and urgency of other therapeutic goals. This information could be used to update the training models and inform how fidelity is measured within systems where adaptation is the norm.

Finally, the current study did not find associations between caregivers’ fidelity and child social communication outcomes. This finding is inconsistent with research demonstrating that child communication outcomes within Project ImPACT are mediated by caregivers’ strategy use (e.g., [12]; [34]). There may be a few reasons that this study did not find this relationship, including the small sample size, the fact of measuring caregiver strategy use within the context of the first and final Project ImPACT session rather than within an isolated play sample, and the narrow data collection window, with caregiver fidelity measured immediately following program completion. Given that caregivers are being served within Part C systems for relatively long periods of time and given that providers are reporting extending their delivery of PMIs to match this context, follow-up data are important.

Although the current study highlights methods that can be used to evaluate associations between routine PMI delivery and a range of therapeutic outcomes, there are several limitations that are important to weigh. First, the ratings of caregiver therapeutic alliance and participatory engagement were not available for every Project ImPACT session or were available for sessions in which a recording of the session was not submitted. This resulted in a sub-sample of only 60 sessions in which to evaluate the association between Project ImPACT delivery and outcomes and resulted in many analyses being insufficiently powered. Second, our observational measure of caregiver fidelity was embedded within the first and final Project ImPACT coaching sessions. While this was done to make data collection more pragmatic within the context of routine EI sessions, it also resulted in a less standardized measure than what is typically used within the context of effectiveness research. Additionally, this study was not a randomized trial, and thus it was not possible to compare how less structured EI service delivery related to the same caregiver and child outcomes. Finally, the current study does not report qualitative data from caregivers and EI providers. Future research is needed to evaluate how EI providers decide to deliver PMIs and, importantly, caregiver preferences regarding the delivery processes.

## 5. Conclusions

In sum, the current study highlights the complex relationship between the delivery of autism PMIs and caregiver outcomes within Part C Early Intervention systems. While providers’ coaching fidelity was positively associated with caregivers’ use of certain Project ImPACT strategies, adaptations that augmented the program appeared to positively affect the caregiver ratings of therapeutic alliance, suggesting that these types of adaptations may result in more responsive delivery practices. The findings highlight the value of holistic provider training models that support providers in maintaining core intervention elements while also being able to effectively tailor manualized PMIs to diverse families and EI service settings.

## Figures and Tables

**Table 1 behavsci-15-01394-t001:** Provider demographic information (N = 21).

	(%) or *Mean* ± *SD*
Demographics	
Age (years)	49.9 ± 12.4
Female Sex	21 (100.0%)
Race	
White	8 (38.1%)
Black or African American	11 (52.4%)
Asian or Pacific Islander	2 (9.5%)
Ethnicity	
Hispanic or Latine	1 (4.8%)
Educational Attainment	
Bachelors	4 (19.1%)
Masters	11 (52.4%)
Doctoral	6 (28.6%)
Work-Related Factors	
Professional Discipline	
Special Instructor (i.e., developmental specialist)	8 (38.1%)
Speech Language Pathologist	4 (19.1%)
Occupational Therapist	3 (14.3%)
Physical Therapist	1 (4.8%)
Board Certified Behavior Analyst	1 (4.8%)
Social Worker	1 (4.8%)
Other	3 (14.3%)
Employment Status (Independently Contracted)	20 (95.2%)
Years of Experience Working within EI Systems	8.6 ± 7.2
Years of Experience Working with Autistic Children	
1–3 years	4 (20.0%)
4–10 years	5 (25.0%)
11–20 years	5 (25.0%)
Greater than 20 years	6 (30.0%)

**Table 2 behavsci-15-01394-t002:** Child and caregiver demographic information (N = 23).

	(%) or *Mean* ± *SD*
Child Characteristics	
Age (months)	24.7 ± 5.7
Female Sex	9 (39.1%)
Race	
White	8 (34.8%)
Black or African American	12 (52.2%)
Asian or Pacific Islander	1 (4.4%)
Multiracial	2 (8.7%)
Ethnicity	
Hispanic or Latine	3 (13%)
Caregiver and Household Characteristics	
Relation to Child—Biological Mother	22 (95.7%)
Race	
White	9 (39.1%)
Black or African American	12 (52.2%)
Asian or Pacific Islander	1 (4.4%)
Multiracial	1 (4.4%)
Ethnicity	
Hispanic or Latine	3 (13%)
Maternal Highest Educational Degree	
High school	6 (26.1%)
Some college/associate	5 (21.7%)
College graduate or higher	12 (52.2%)
Household income	
Low income (USD 39,999 or less)	8 (34.8%)
Middle income (USD 40,000–USD 74,999)	5 (21.7%)
High income (USD 75,000 or more)	10 (43.5%)

**Table 3 behavsci-15-01394-t003:** Schedule of measures across study time period.

	*t* _1_	*t* _2_	*t* _3_
**PPROVIDER MEASURES**
Demographic Information	X		
Project ImPACT Coaching Fidelity		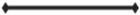
Project ImPACT Adaptation		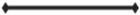
**CAREGIVER MEASURES**
Demographic Information		X	
Parent Observation of Social Interaction (POSI)		X	
Session Participatory Engagement		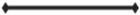
Session Therapeutic Alliance		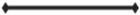
Family Outcome Scale, Revised (FOS-R)		X	X
Caregiver Project ImPACT Fidelity		X	X
Social Communication Checklist (SCC)		X	X

*Note*: *t*_1_: Pre-Project ImPACT training (providers); *t*_2_: Pre-Project ImPACT delivery; *t*_3_: Post-Project ImPACT delivery.

**Table 4 behavsci-15-01394-t004:** Repeated measures correlations between coaching fidelity and outcomes across 60 sessions.

**Independent Variable**	**Dependent Variable**	**Repeated Measures** ***r* (95% CI)**	***p*-Value**
Coaching Fidelity	Therapeutic alliance	0.00 (−0.36, 0.35)	0.992
Coaching Fidelity	Participatory engagement	−0.24 (−0.59, 0.11)	0.180

Repeated measures correlations, 95% CIs, and *p*-values were estimated using linear mixed models as recommended by [8] ([8]), accounting for variation between dyads via random effects and clustering of repeated sessions within each dyad.

**Table 5 behavsci-15-01394-t005:** Repeated measures comparison of therapeutic alliance and participatory engagement across sessions within and without augmenting and reducing adaptations.

	Sessions Without Augmenting Adaptations	Sessions with One or More Augmenting Adaptations	*p*-Value	ES
Repeated Measures, LS *Mean* (95% CI)				
Therapeutic alliance	4.73 (4.56, 4.89)	4.94 (4.80, 5.00)	**0.048**	0.63
Participatory engagement	4.51 (4.23, 4.79)	4.36 (3.45, 5.00)	0.676	0.23
	**Sessions without a reducing adaptation**	**Sessions with one or more reducing adaptations**	***p*-Value**	**ES**
**Repeated Measures, LS *Mean* (95% CI)**				
Therapeutic alliance	4.74 (4.55, 4.92)	4.83 (4.67, 4.99)	0.445	0.23
Participatory engagement	4.45 (4.16, 4.74)	4.33 (3.82, 4.84)	0.679	0.16

Upper 95% confidence limits truncated at 5 due to the measure’s scale range. Repeated measures results were estimated using linear mixed models that accounted for clustering of repeated sessions within each provider–caregiver dyad, modeled separately for sessions with and without augmenting or reducing adaptations. Effect size (ES) was calculated by dividing the least squares mean difference by the square root of the average variance estimate from the mixed models and interpreted using Cohen’s *d* criteria: small (0.2), moderate (0.5), and large (0.8).

**Table 6 behavsci-15-01394-t006:** Paired differences in caregiver empowerment, caregiver fidelity, and child social communication skills from Project ImPACT start (Time 1) to completion (Time 2) for 18 caregiver–child dyads.

Measure	Time 1 *Mean* (*SD*)	Time 2 *Mean* (*SD*)	Paired *p*-Value	ES
Family Outcomes Scale				
Understanding Strength and Needs	14.72 (3.58)	16.94 (2.18)	**0.035**	0.54
Knowing Your Rights	17.44 (5.28)	19.33 (4.52)	0.258	0.28
Helping Your Child Develop and Learn	15.39 (3.65)	17.61 (2.38)	**0.021**	0.60
Support Systems	19.44 (4.45)	20.94 (3.23)	0.212	0.31

Caregiver Fidelity				
Focus on Child	3.13 (0.89)	3.44 (0.96)	0.206	0.33
Adjust Communication	2.63 (0.89)	3.13 (0.81)	**0.041**	0.56
Create Opportunities	2.06 (0.93)	2.50 (0.97)	0.090	0.46
Teach New Skills	1.63 (0.62)	2.25 (0.86)	**0.020**	0.65
Shape the Interaction	2.19 (0.75)	2.88 (0.89)	**0.003**	0.87

Social Communication Checklist				
Social Engagement	34.05 (6.86)	38.68 (5.40)	**<0.001**	1.02
Expressive Communication	22.44 (5.91)	24.67 (5.60)	**0.004**	0.79
Receptive Communication	14.17 (4.53)	16.33 (5.12)	**0.003**	0.82
Imitation	11.00 (3.48)	12.47 (3.52)	**0.017**	0.60
Play	17.82 (5.17)	20.06 (4.72)	**0.028**	0.59

Paired *p*-values were estimated using paired *t*-tests. Effect size (ES) was calculated by dividing the mean paired difference by the standard deviation of the paired differences for each variable and interpreted using Cohen’s *d* criteria: small (0.2), moderate (0.5), and large (0.8).

**Table 7 behavsci-15-01394-t007:** Correlation results evaluating the strength and direction of the relationships between Time 2 outcomes and mean coaching fidelity when controlling for Time 1 for 18 caregiver–child dyads.

Outcome	Mean Coaching Session Fidelity Partial Pearson r (95% CI) ^1^	*p*-Value
Family Outcomes Scale		
Understanding Strength and Needs	−0.17 (−0.60, 0.34)	0.504
Knowing Your Rights	−0.19 (−0.62, 0.32)	0.452
Helping Your Child Develop and Learn	−0.46 (−0.77, 0.03)	0.056
Support Systems	−0.39 (−0.74, 0.11)	0.109
Caregiver Fidelity		
Focus on Child	0.17 (−0.38, 0.63)	0.546
Adjust Communication	−0.14 (−0.61, 0.40)	0.623
Create Opportunities	0.33 (−0.22, 0.72)	0.223
Teach New Skills	0.39 (−0.16, 0.75)	0.142
Shape the Interaction	0.19 (−0.36, 0.64)	0.487
Social Communication Checklist		
Social Engagement	−0.31 (−0.68, 0.19)	0.206
Expressive Communication	−0.16 (−0.59, 0.35)	0.533
Receptive Communication	0.19 (−0.32, 0.61)	0.467
Imitation	0.02 (−0.45, 0.48)	0.942
Play	−0.31 (−0.70, 0.22)	0.231

^1^ Partial Pearson correlations evaluated associations between Time 2 outcomes and coaching fidelity, controlling for Time 1 scores, among 18 caregiver–child dyads. Mean coaching fidelity was computed by averaging fidelity scores across multiple sessions for each provider.

**Table 8 behavsci-15-01394-t008:** Repeated measures correlations between caregiver fidelity and social communication skills for 18 caregiver–child dyads.

Independent Variable Caregiver Fidelity	Dependent Variable SCC Sub-Scales	Repeated Measures *r* (95% CI)	*p*-Value
Developmental Strategies Overall	SCC Engagement	−0.14 (−0.51, 0.22)	0.448
	Expressive Communication	−0.08 (−0.46, 0.30)	0.696
	Understand Communication	0.07 (−0.32, 0.45)	0.731
Behavioral Strategies Overall	SCC Engagement	−0.07 (−0.42, 0.28)	0.682
	Expressive Communication	−0.12 (−0.46, 0.22)	0.497
	Understand Communication	−0.18 (−0.54, 0.17)	0.314
Shape Overall	SCC Engagement	−0.06 (−0.43, 0.31)	0.751
	Expressive Communication	−0.18 (−0.54, 0.17)	0.312
	Understand Communication	−0.16 (−0.54, 0.22)	0.405

Repeated measures correlations, 95% CIs, and *p*-values were estimated using linear mixed models as recommended by [8] ([8]), accounting for variation between dyads via random effects and clustering of repeated sessions within each dyad.

## Data Availability

The data included in the current manuscript can be obtained by contacting the corresponding author at katherine.e.pickard@emory.edu.

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
