# Peer review of "Characterizing the Relationship Between Intervention Delivery and Outcomes Within Part C Community Settings"

_behavsci, 2025, doi:10.3390/bs15101394_

Round 1

Reviewer 1 Report

Comments and Suggestions for Authors

The manuscript represents an important contribution to the field of early intervention. Overall this is a well written manuscript that offers clear and important future research and clinical directions. There are only a few major areas that should be addressed prior to publication:

  1. Alignment of research questions and analysis for 3.2. This question asks about the association between fidelity and adaptation (IVs) and caregiver self-efficacy, caregiver use of intervention strategies, and child social communication (DVs). However it is unclear why Table 6 is included (which is depicting changes in these caregiver and child SC outcomes over time). Furthermore, the results section exclusively focus on fidelity (Table 7) without any reporting of results for intervention adaptations. Additionally, Table 8 does not provide any additional information related to the research question, but does raise question about spurious results given the number of correlations run for such a small sample.
  2. Multiple Testing: Given the size of the sample coupled with the number of statistical tests performed, there is some concern about the likelihood that some findings may be due to chance. This should either be addressed in the discussion section as a limitation or in the analysis plan by adjusting p values. 
  3. Discussion of Results: statements such as "caregivers demonstrated significant increased in their self efficacy" are not only not related to the research questions, but they are also not justified by the study design (pre post).
  4. Inverse relation between fidelity and self efficacy: consider that perhaps the caregiver self efficacy might have had an influence on coaching fidelity (versus the opposite direction which was considered). For example, it could be that coaches could see a parent with lower self efficacy and therefore knew that they needed to demonstrate all of the coaching behaviors versus another coach working with a caregiver who is demonstrating high self efficacy, and the coach is able to be more lax with the implementation of coaching. 
  5. Consideration of Alliance : Alliance is likely also related to self efficacy yet that association was not explored in the paper. Consider examining the extent to which alliance impacts self-efficacy especially in light of the inverse findings related to coaching fidelity and self-efficacy. 

I believe that all of my concerns are easily addressable and that this paper has the potential to be very impactful. 

Author Response

Point 1: This question asks about the association between fidelity and adaptation (IVs) and caregiver self-efficacy, caregiver use of intervention strategies, and child social communication (DVs). However it is unclear why Table 6 is included (which is depicting changes in these caregiver and child SC outcomes over time). Furthermore, the results section exclusively focus on fidelity (Table 7) without any reporting of results for intervention adaptations. Additionally, Table 8 does not provide any additional information related to the research question, but does raise question about spurious results given the number of correlations run for such a small sample.

Response 1: One of the aims of this manuscript is: How do providers’ Project ImPACT fidelity and adaptation delivery affect changes in: (a) caregiver self-efficacy; (b) caregivers’ use of Project ImPACT strategies; and (c) child social communication outcomes. We believe that it is important to report on caregiver overall changes on the DVs prior to reporting whether fidelity and adaptation relate to these variables. Thus, we report on changes in these constructs before reporting on the relationship to intervention fidelity. We report on the relationship between adaptation and measures of caregiver engagement and therapeutic alliance in the results section. Because we do not anticipate that adaption will affect child social communication outcomes or caregiver fidelity, those relationships were not explored.

Point 2: Given the size of the sample coupled with the number of statistical tests performed, there is some concern about the likelihood that some findings may be due to chance. This should either be addressed in the discussion section as a limitation or in the analysis plan by adjusting p values. 

Response 2: We appreciate this thoughtful comment. The study involved multiple distinct outcomes, and given the exploratory nature of the work in a small community-based sample, we chose not to apply formal multiplicity corrections. We recognize that this approach increases the risk of Type I error, but felt it was important not to over-correct and potentially miss meaningful signals in the data. We have added a statement to the Discussion noting this limitation. We also emphasize effect sizes in the Results and Discussion to highlight the practical significance of the findings beyond p-values.

Point 3: Inverse relation between fidelity and self efficacy: consider that perhaps the caregiver self efficacy might have had an influence on coaching fidelity (versus the opposite direction which was considered). For example, it could be that coaches could see a parent with lower self efficacy and therefore knew that they needed to demonstrate all of the coaching behaviors versus another coach working with a caregiver who is demonstrating high self efficacy, and the coach is able to be more lax with the implementation of coaching. 

Response 3: We appreciate this point and have added a statement to the discussion that considers this possibility. 

Point 4: Consideration of Alliance : Alliance is likely also related to self efficacy yet that association was not explored in the paper. Consider examining the extent to which alliance impacts self-efficacy especially in light of the inverse findings related to coaching fidelity and self-efficacy. 

Response 4: While we appreciate this suggestion, our main aims are to examine the relationship between Project ImPACT delivery and caregiver outcomes. Examining the relationship between caregiver-rated therapeutic alliance and self-efficacy is not within the study aims. 

Reviewer 2 Report

Comments and Suggestions for Authors

Thank you for the opportunity to review the manuscript titled "Characterizing the relationship between intervention delivery and outcomes within Part C community settings". The authors address an area of research that is sorely needed: the case of PMIs, intervention fidelity, and the level of support required. The findings are very interesting and well presented. However, I can’t comment on the accuracy of the methodology because I am not familiar with linear mixed modeling. I will list my suggestions below:

  • Provide a definition for therapeutic allegiance. This concept is introduced in the abstract and mentioned in the introduction, but it is not explained until page 7, where it is only briefly discussed.
  • Please indicate if the participants received any incentives for their participation in this study.
  • In Table 1 clarify what you mean by “special instructor” is this a trained instructor? A teacher of students with disabilities? Some other type of professional?
  • In Table 2, given the numbers, it seems that Hispanic/Latine is categorized as ethnicity, not race. This should be clarified in the table.
  • The measures in Table 3 would be more effective if they were presented in the same order in the narrative. Additionally, there is no description of the “POSI” measure in the narrative.
  • Reliability data is missing for the “project impact adaptation” measure.
  • For the “coaching fidelity” measure, reliability was only calculated for 20% of the sessions, yielding an ICC of 0.77, which is on the lower end of what is acceptable. I suggest the authors provide reasoning behind these results.
  • On page 10, lines 247-356 would be better suited in the discussion section of the paper.
  • There are several instances where the authors mention the use of “fine-grained data collection”, but they do not explain how it is different than other methods.

Author Response

Reviewer 2:

Point 1: Provide a definition for therapeutic allegiance. This concept is introduced in the abstract and mentioned in the introduction, but it is not explained until page 7, where it is only briefly discussed.

Response 1: Thank you for pointing this out. We have now provided a brief definition of therapeutic alliance in the introduction. 

Point 2: Please indicate if the participants received any incentives for their participation in this study.

Response 2: We have now clarified whether caregivers received compensation as part of the methods section.

Point 3: In Table 2, given the numbers, it seems that Hispanic/Latine is categorized as ethnicity, not race. This should be clarified in the table.

Response 3: We apologize for ethnicity getting edited out. We have added it back into the demographic tables. 

Point 4: There is no description of the “POSI” measure in the narrative.

Response 4: The POSI is now described within the methods section.

Point 5: Reliability data is missing for the “project impact adaptation” measure.

Response 5: Thank you for pointing this out. Reliability information has been added for the adaptation measure. 

Point 6: On page 10, lines 247-356 would be better suited in the discussion section of the paper.

Response 6: In the sections you are refernecing, we see descriptions of the results (i.e., relationships between fidelity and self-efficacy). If this is what is being referenced, we believe that the statistical findings and table belong in the results section. 

Point 7: There are several instances where the authors mention the use of “fine-grained data collection”, but they do not explain how it is different than other methods.

Response 7: We agree that the term fine-grained was not defined. We have added definition in the introduction and have removed that term in other places.